# LRP-1 Matricellular Receptor Involvement in Triple Negative Breast Cancer Tumor Angiogenesis

**DOI:** 10.3390/biomedicines9101430

**Published:** 2021-10-09

**Authors:** Océane Campion, Jessica Thevenard Devy, Clotilde Billottet, Christophe Schneider, Nicolas Etique, Jean-William Dupuy, Anne-Aurélie Raymond, Camille Boulagnon Rombi, Marie Meunier, El-Hadi Djermoune, Elodie Lelièvre, Amandine Wahart, Camille Bour, Cathy Hachet, Stefano Cairo, Andréas Bikfalvi, Stéphane Dedieu, Jérôme Devy

**Affiliations:** 1UFR Sciences Exactes et Naturelles, Université de Reims Champagne-Ardenne, 51687 Reims, France; campionoceane@gmail.com (O.C.); jessica.devy@univ-reims.fr (J.T.D.); christophe.schneider@univ-reims.fr (C.S.); nicolas.etique@univ-reims.fr (N.E.); marie.meunier@givaudan.com (M.M.); elodie.lelievre4@gmail.com (E.L.); amandine.wahart@gmail.com (A.W.); camille.bour@univ-reims.fr (C.B.); cathy.hachet@univ-reims.fr (C.H.); stephane.dedieu@univ-reims.fr (S.D.); 2Matrice Extracellulaire et Dynamique Cellulaire, MEDyC, UMR 7369 CNRS, 51687 Reims, France; camille.boulagnon@gmail.com; 3INSERM, LAMC, U1029, Université de Bordeaux, 33600 Pessac, France; clotilde.billottet@u-bordeaux.fr (C.B.); andrea.bikfalvi@u-bordeaux.fr (A.B.); 4Plateforme Protéome, Université de Bordeaux, 33076 Bordeaux, France; jean-william.dupuy@u-bordeaux.fr; 5Plateforme Oncoprot, TBM-Core US 005, 33000 Bordeaux, France; anne-aurelie.raymond@inserm.fr; 6Laboratoire d’Anatomie Pathologie, CHU Reims, 51100 Reims, France; 7CNRS, CRAN, Université de Lorraine, 54000 Nancy, France; el-hadi.djermoune@univ-lorraine.fr; 8Xentech, 91000 Evry-Courcouronnes, France; Stefano.Cairo@xentech.eu

**Keywords:** breast cancer, TNBC, LRP-1, angiogenesis

## Abstract

Background: LRP-1 is a multifunctional scavenger receptor belonging to the LDLR family. Due to its capacity to control pericellular levels of various growth factors and proteases, LRP-1 plays a crucial role in membrane proteome dynamics, which appears decisive for tumor progression. Methods: LRP-1 involvement in a TNBC model was assessed using an RNA interference strategy in MDA-MB-231 cells. In vivo, tumorigenic and angiogenic effects of LRP-1-repressed cells were evaluated using an orthotopic xenograft model and two angiogenic assays (Matrigel^®^ plugs, CAM). DCE-MRI, FMT, and IHC were used to complete a tumor longitudinal follow-up and obtain morphological and functional vascular information. In vitro, HUVECs’ angiogenic potential was evaluated using a tumor secretome, subjected to a proteomic analysis to highlight LRP-1-dependant signaling pathways. Results: LRP-1 repression in MDA-MB-231 tumors led to a 60% growth delay because of, inter alia, morphological and functional vascular differences, confirmed by angiogenic models. In vitro, the LRP-1-repressed cells secretome restrained HUVECs’ angiogenic capabilities. A proteomics analysis revealed that LRP-1 supports tumor growth and angiogenesis by regulating TGF-β signaling and plasminogen/plasmin system. Conclusions: LRP-1, by its wide spectrum of interactions, emerges as an important matricellular player in the control of cancer-signaling events such as angiogenesis, by supporting tumor vascular morphology and functionality.

## 1. Introduction

Breast cancer (BC) is the most diagnosed cancer in women worldwide and the leading cause of cancer-related death. It is an heterogenous disease characterized by diverse phenotypes and a considerable heterogeneity in molecular and histopathological features [1]. Based on transcriptomics analysis, five BC subtypes have been identified: luminal A, luminal B and human epidermal growth factor 2 receptor (HER2)—enriched, basal-like, and normal-like [2,3,4]. From a morphological perspective, BC subtypes are discriminated according to histological observations, tumor grade, lymph nodes, and predictive immunohistochemistry markers detection such as estrogen and progesterone receptors (ER and PR) or HER2. Triple negative breast cancer (TNBC), accounting for approximately 15% of all BC, is characterized by the non-expression of ER, PR, and the lack of HER2 overexpression and/or amplification [2] associated with mesenchymal characteristics as well as a stem cell-like expression signature [5]. As the basal-like tumor subtype does not usually express ER, PR, and HER2, they tend to be referenced as TNBC. However, by the identification of gene expression, these tumors are distinct, although not mutually exclusive [6,7]. It has been showed in a cohort that 77% of basal-like tumors were TNBC, while 71% of TNBC were basal-like [7]. In the absence of hormonal receptors and HER-2 receptor expression, TNBC patients do not benefit from the currently available receptor-targeted systemic therapies, such as hormonal and trastuzumab-based therapies. Thus, TNBC, remaining refractory to targeted and conventional therapy advancements [3], requires the identification of novel therapeutic targets in order to enhance the therapeutic options. In recent years, the tumor microenvironment research has occupied an important place in the cancer research field [8]. It is widely recognized that the primary cancer invasiveness is determined not only by the tumor cells’ genotype and phenotype, but also by their interactions with the extracellular environment, variously composed of cellular types, which modulate tumor development and infiltration capacities as well as angiogenic responses [4]. Once a tumor lesion increases a few millimeters in diameter, hypoxia and nutrient deprivation trigger the “angiogenic switch” to allow tumor progression [9]. Tumor cells exploit their microenvironment by releasing soluble mediators such as growth factors, chemokines, and cytokines to activate normal, quiescent cells around them and initiate a cascade of events that quickly defects. The magnitude and quality of the angiogenic response are ultimately determined by the balance of pro- and anti-angiogenic signals and, more specifically, their unique activities on multiple cell types [10]. There are several classical or more sophisticated mechanisms leading to the formation of new vessels within a tumor. Among them, there are sprouting and intussusceptive ngiogenesis, co-option of preexisting vessels, vascular mimicry, or angiogenesis from endothelial stem cells [11]. All these mechanisms are available to serve the tumor’s exacerbated need to survive, proliferate, and invade adjacent tissues. The endocytic low-density lipoprotein receptor-related protein-1 (LRP-1) plays a crucial role in controlling membrane proteome dynamics [12,13]. This receptor is composed of a 515 kDa α extracellular chain containing extracellular ligand-binding domains organized in four clusters and an 85 kDa transmembrane β-chain containing a short cytoplasmic domain characterized by 2 NPxY motifs (Asn-Proline-X-Tyrosine) triggering endocytosis. LRP-1 directly participates in the extracellular matrix (ECM) remodeling through the endocytosis of numerous active proteinases or proteinase/inhibitor complexes [14]. LRP-1 is also involved in cell migration, a key process in the acquisition of tumor cell invasiveness, by modulating integrin functions through a subtle control of their endocytosis/recycling cycle [15]. In addition to its catabolic activity, LRP-1 binds to many proteins via its intracytoplasmic chain (ICD) to constitute a cellular signaling platform. By activating the MEK/ERK pathway and concomitantly inhibiting MKK7/JNK, LRP-1 can regulate cancer cells anchoring to the ECM [16]. A previous anatomopathological study has revealed that LRP-1 is predominantly overexpressed in TNBC and HER2^+^ BC compared to other subtypes [17]. This prompted us to take an interest in its involvement in TNBC progression and, more specifically, in its direct or indirect role in controlling the angiogenic balance and whether this role could be decisive for tumor progression [17,18,19,20]. The present work investigates LRP-1’s tumorigenic potential in MDA-MB-231 cells using an RNA interference strategy. By means of an orthotopic mammary fat pad model, a Matrigel ™ plug, and a CAMs assay, we showed that LRP-1 repression affects in vivo tumor growth by modulating, inter alia, angiogenesis. Further functional studies based on in vitro tumor conditioned media (TCM) effects on human umbilical vein endothelial cells’ (HUVECs) behaviors and angiogenic capabilities reinforced LRP-1’s role in modulating the angiogenesis process. A proteomics analysis of TCM showed that LRP-1 supports angiogenesis and tumor growth through the TGF-β signaling and plasminogen/plasmin system modulation, among others. Thus, the matricellular receptor LRP-1, by its wide spectrum of interactions within the microenvironment, appears as a key factor in the control of BC signaling events such as angiogenesis.

## 2. Materials and Methods

### 2.1. Cell Lines and Culture

MDA-MB-231 breast cancer cells were purchased from DSMZ (Braunschweig, Germany) and grown in 10% fetal bovine serum (FBS) (ATCC^®^ 30-2020™, LGC, Molsheim, France) DMEM (1 g/L of glucose). The cells were stably transfected with the pOPRSV-1/MCS expression plasmid (Agilent Technologies, Courtaboeuf, Les Ulis, France), containing a coding LRP-1-interfering RNA sequence: AAG CAG TT GCC TGC AGA GAT (shLRP-1) or a scramble control sequence: CCA GTC GCC ATTA ATT ATG CAA (shCtrl). We used 0.8 mg/mL of geneticin (G418; Sigma Aldrich, St. Louis, MI, USA) for the selection. For the screening needs, MCF-7, T-47D, SK-Br3, Hs-578T, BT-20, and 4T1 breast cancer cell lines have also been used. HUVECs (C-12203) were purchased from Promocell (Handschuhsheimer, Germany) and HUVECs-GFP labeled with GFP (ZHC-2402) from Cellworks (Buckingham, UK). The cells were cultivated in EBM^TM^-2 Basal Medium (CC-3156) supplemented with EGM^TM^-2 SingleQuots^TM^ (CC-4176; Lonza, Walkersville, MD, USA). The cells at the passage from P3 to P5 were used.

For all procedures, the cells were harvested using a 1X Trypsin-EDTA solution (Sigma Aldrich, St. Louis, MI, USA) or Accutase^®^ (C-41310; Promocell, Handschuhsheimer, Germany) and were maintained at 37 °C in a humidified atmosphere of 5% CO_2_.

### 2.2. Tumor- and HUVEC-Conditioned Media Preparation

48-h tumor-conditioned media (TCM): shLRP-1 or shCtrl MDA-MB-231 cells were seeded at 3.2 × 10^6^ in T150 culture flasks. Forty-eight hours after seeding, the media were replaced by 8 mL of DMEM containing 1% FBS. After 48 h of incubation, the supernatant was centrifuged at 10,000× *g* for 10 min. In parallel, cells were harvested and counted using Scepter^TM^ 2.0 (Merck Millipore, Molsheim, France). Cell equivalents between shLRP1 and shCtrl TCM were made by diluting the most concentrated TCM with DMEM. The resulting TCM, equivalent in pairs at a cell concentration from 0.8 to 1.2 million cells/mL, were stored in aliquots at 20 °C to avoid multiple freeze–thaws.

24-h TCM: shLRP-1 or shCtrl MDA-MB-231 cells were seeded at 1.2 × 10^6^ in a 35-mm culture dish. Forty-eight hours after seeding, the media were replaced by 3.5 mL of FBS-free, phenol-red-free DMEM after washing the cells with PBS twice. After 24 h of incubation, the supernatant was centrifuged at 10,000× *g* for 10 min. In parallel, cells were detached and counted using Scepter^TM^ 2.0 (Merck Millipore, Molsheim, France). Cell equivalents between shLRP-1 and shCtrl TCM were made by diluting the most concentrated TCM in DMEM. The resulting TCM, equivalent in pairs at a cell concentration from 0.8 to 1.2 million cells/mL, were stored in aliquots at 20 °C to avoid multiple freeze–thaws.

24-h TCM-stimulated HUVEC-conditioned medium (CM): HUVECs were seeded at 1.2 × 10^6^ in a 35-mm culture dish. Twenty-four hours after seeding, the media were replaced by 24 h of shLRP-1 or shCtrl MDA-MB-231 TCM as a pre-treatment for 24 h after washing the cells with PBS twice. After treatment incubation, the media were replaced by 3.5 mL of FBS-free, phenol-red-free DMEM after washing the cells twice with PBS. After 24 h of incubation, the supernatant was centrifuged at 10,000× *g* for 10 min. The resulting CMs were stored in aliquots at 20 °C to avoid multiple freeze–thaws.

### 2.3. In Vivo Studies 

Mice (5–6 week-old female Balb/c nu) purchased from Janvier (Janvier labs, Le Gnest-Saint-Isle, France) were housed in ventilated cages under filtered air and acclimatized for one week prior to manipulation. The experiments with animals were approved and carried out in compliance with ethics rules under the authorization number APAFIS#4373-2016030410575189 vI, “Study of LRP-1 receptor involvement in TNBC models in mice”, distributed by the higher education and research administration attached to the French National Education Ministry. All procedures were conducted under general anesthesia induced by the inhalation of 3% isoflurane and maintained with 1.5% during imaging. 

### 2.4. Orthotopic Xenograft Model 

shLRP-1 or shCtrl MDA-MB-231 cells were harvested using Accutase^®^, washed and resuspended into a 5 × 10^7^/mL cell solution before inoculation. Twelve mice were injected with 100 µL into the mammary fat pad. Tumor growth was assessed by measuring the length (A) and width (B) with a digital caliper every week. The volumes were calculated using 1/2(A × B^2^). The mice were sacrificed 28 days after inoculation. After excision, the tumor tissues were immersed in liquid nitrogen, transferred to a vial, and stocked at −80 °C or fixed in 4% paraformaldehyde (Sigma Aldrich, Saint-Louis, MI, USA) for 24 h and embedded in paraffin. 

### 2.5. Matrigel^®^ Plug

A total of 2 × 10^5^ of shLRP-1 or shCtrl MDA-MB-231 cells were resuspended in 0.1 mL of growth medium, mixed with 0.4 mL of growth factor-reduced Matrigel^®^ (Corning^®^, BD Biosciences, Franklin Lakes, NJ, USA) at 8.6 mg/mL, and implanted subcutaneously into the flank of each 7-week-old female BALB/c-nu mouse (Janvier labs, Le Genest-Saint-Isle, France) (*n* = 12/group). Twenty-one days after the injection, the animals were sacrificed, and the Matrigel^®^ plugs were excised, photographed, and fixed in 4% paraformaldehyde (Sigma Aldrich, Saint-Louis, NJ, USA) for histological analysis. 

### 2.6. Optical Imaging 

Fluorescent molecular tomography (FMT) was conducted using an FMT-4000 scanner (PerkinElmer, Waltham, MA, USA) calibrated beforehand with fluorophores according to the supplier’s instructions. Fluorescence quantification was achieved with the TrueQuant 3.0 software (PerkinElmer, Waltham, MA, USA). The AngioSense^TM^-750/AngioSense^TM^-680 or HypoxiSense^TM^-680 contrast agent was used, with an excitation filter of 750 ± 3 nm or 670 ± 3 nm, respectively, and an emission filter of 690–740. 3D and 2D trans-illumination acquisitions were carried out 24 h after the injection of a 100 µL intravenous contrast agent.

### 2.7. MRI Imaging 

Images were acquired using a three-dimensional (3D) coronal T2-weighted fast spin echo sequence (FSE) with echo time (TE)/repetition time (TR) = 68/5000 ms, a 60 mm × 60 mm field of view, 1 mm slice thickness, and a 512 × 240 matrix (with 2× oversampling in the read direction) zero-padded to 512 × 256 during reconstruction (resolution = 0.234 mm × 0.25 mm), echo train length = 8, number of averages = 1, and an effective receiver bandwidth (BW) = 20 kHz); a transverse T2-weighted fast spin echo sequence (FSE) with echo time (TE)/repetition time (TR) = 68/5000 ms, a 40 mm × 40 mm field of view, 1 mm slice thickness, and a 256 × 240 matrix zero-padded to 256 × 256 during reconstruction (resolution = 0.156 mm × 0.167 mm), echo train length = 8, number of averages = 2, receiver bandwidth (BW) = 20 kHz); with echo time (TE)/repetition time (TR) = 11/720 ms, a 16 mm × 16 mm × 16 mm field of view, a 128 × 128 × 80 matrix (resolution = 0.125 mm × 0.125 mm × 0.2 mm, echo train length = 4, number of average = 2, flip angle = 40°, bandwidth (BW) = 100 kHz), and a T1 fast spin echo sequence (FSE) with echo time (TE)/repetition time (TR) = 60/5000 ms, a 16 mm × 16 mm × 16 mm field of view, and a 128 × 128 × 80 matrix (resolution = 0.125 mm × 0.125 mm × 0.2 mm, echo train length = 4, number of averages = 2, flip angle = 40°, bandwidth (BW) = 100 kHz). Perfusion was measured in DCE-MRI (TR = 60 ms, flip angle = 25° and 105 experiments) by an intravenous bolus injection of Clariscan^TM^ (gadopentate, dimeglumine, Bayer Healthcare, Bayergadoteric acid, GE Healthcare), 100 μL and 0.2 mmol/Kg at experiments 6/105. The total imaging time was ~40 min. Intensity analyses were performed after a 4D reconstruction with the OsiriX software (Pixmeo, Swiss) using the ROI enhancement plugin. The intensity of the ROI in the plugs was quantified with the OsiriX software after the 4D reconstruction. A mouse body coil was used. The temperature was maintained by a heating air flow built into the animal bed system. Respiration was monitored throughout the entire scan. 

### 2.8. Chick Chorioallantoic Membrane (CAM) Assay 

Fertilized chicken eggs (EARL Les Bruyères, FR) (less sentient model according to 3R rule) were incubated in egg-racks at 37 °C and a 65% humidified atmosphere in a rotatory incubator allowing rotations of 25 degrees every 6 h. On day 3 of embryonic development, a window was made in the eggshell and sealed with adhesive film (Durapore tape). On embryonic day 10, once the CAM was fully formed, a gentle laceration of its surface was performed using a scalpel, and plastic rings (made from Nunc Thermanox coverslips) were put on the surface of the CAM [21]. Then, 4 × 106 of shLRP-1 or shCtrl MDA MB-231 cells were deposited as a thin layer on the surface of the fertilized chicken eggs. On day 17, CAMs were fixed in vivo (4% paraformaldehyde, RT, 30 min) and included. Digital photographs were taken under a Nikon SMZ800 stereomicroscope. A quantitative analysis was performed using a MATLAB routine developed by Dr El-Hadi Djermoune [22] in which the vascular structures’ segmentation leaned on the vesselness probability map of the 2D images [23].

### 2.9. Histology

A histological analysis of formaldehyde-fixed and paraffin-embedded tumors and Matrigel^®^ plugs was performed on 4 μm hematein, eosin, and saffron (HES)-stained sections prepared using routine methods. The necrosis area was calculated as the ratio of (necrosis surface/tumor surface) × 100 and the mitotic index in 10 consecutive high-power fields (HPF = 0.31 mm^2^) in the highest mitotic activity area.

### 2.10. Immunohistochemistry and Immunofluorescence

For immunohistochemistry, tumor and Matrigel^®^ plug sections were deparaffinized in xylene and rehydrated in solutions of graded ethanol. The antigen retrieval was performed at 95 °C for 40 min in 0.01 M sodium citrate buffer sections, which were then immersed in 3% hydrogen peroxide for 30 min to block the endogenous peroxidase activity. For vessel labeling, the sections were incubated with an anti-CD31 [EPR17259] (1/100, rabbit monoclonal, ab18298; Abcam, Cambridge, UK) primary antibody (1/100) at 4 °C overnight. Following primary antibody incubation and washing, the sections were treated using the labeled polymer peroxidase AEC method (Dako EnVisionThermo ScientificTM HRP Lab VisionTM RTU AEC Substrate System Kit, DakoFisher Scientific, CarpinteriaPittsburgh, CAPA) for 60 min. The proteins were visualized using a liquid diaminobenzidine substrate kit (Zymed Laboratories, San Francisco, CA, USA). The sections were counterstained with hematoxylin before mounting. Appropriate positive and negative controls were used throughout the experiment. CD31-immunostained vessels, exhibiting a lumen, were numerated on 5 consecutive HPF in the area with the vessels’ greatest density. The interpretation was fulfilled blindly by an external anatomopathologist. 

For immunofluorescence, an anti-vimentin (V6) (1/100, mouse monoclonal, sc-6260; Santa Cruz, CA, USA) primary antibody was used. For the dilution of the antibodies, the preincubation and washing of the cryosections, 0.4% Triton X-100, and 1% bovine serum albumin diluted in a potassium phosphate buffer (pH 7.4) were performed. All processing of cryosections was done in humidified chambers. The primary antibody was applied overnight at 4 °C. After four washes within 20 min, the secondary antibody was applied for 60 min. After repeated washing, the sections were mounted under coverslips with ProLong^TM^ Gold Antifade Mountant (Invitrogen, Waltham, MA, USA). Appropriate positive and negative controls were used throughout experiments.

### 2.11. Protein Extraction and Western Blot (WB) Analysis 

WB was realized as described in [19], using the anti-LRP-1 β-chain [EPR3724] (1/10^4^, rabbit monoclonal, ab92544; Abcam, Cambridge, UK).

### 2.12. RNA Isolation and Real-Time PCR 

Total mRNA was extracted using the TRIzol reagent (Thermo Fisher Scientific, Waltham, MA, USA). PCR primers were synthesized by Eurogentec (Liege, Belgium) as follows (5′-3′) for LRP1: GCTATCGACGCCCCTAAGAC and CGCCAGCCCTTTGAGATACA (Appendix A). mRNA analyses were performed on breast cancer cell lines. The total RNA was isolated using Extract-All (Eurobio, Les Ulis, France) and DNase treated (RQ1 RNase-Free DNase, Promega) as described in the manufacturer’s instructions. The RNA quality was checked by 1% agarose gel electrophoresis, and the total RNA concentration (ng/μL) was measured at 260 nm by a NanoDropTM One (Thermo Fisher Scientific, Waltham, MA, USA) for each sample. Two hundred and fifty nanograms of total RNA were used for reverse-transcription using the VERSO cDNA kit (Thermo Fisher Scientific, Waltham, MA, USA) according to the manufacturer’s instructions, using a mix of random hexamer primers and anchored oligo dT. The transcript levels were determined by a real-time quantitative analysis using an Absolute SYBR Green Rox mix (Fisher Scientific) on a CFX 96 touch real time PCR detection system (Bio-Rad). PCR reactions were carried out in duplicates in 96-well plates (15 μL per well) in a buffer containing 1× SYBR Green mix (including Taq polymerase, dNTPs, SYBR Green dye), 280 nM forward and reverse primers, and a 1:10 dilution of reverse transcript RNA. After denaturation at 95 °C for 15 min, the amplification occurred in a two-step procedure: 10 s of denaturation at 95 °C and 45 s of annealing/extension at 60 °C, with a total of 40 cycles. Identical thermal cycling conditions were used for all targets. The specificity of PCR amplification was checked using a heat dissociation curve from 65 °C to 95 °C following the final cycle. The cycle threshold (Ct) values were recorded with the Bio-Rad CFX ManagerTM 3.1 software (Bio-Rad). Specific primers were designed using the Primer3 and BLAST softwares (National Center for Biotechnology Information) and are presented in the Appendix A. The PCR efficiency of the primer sets was calculated by performing a real-time PCR on serial dilutions and was 90% to 110%. For each experiment, PCR reactions were performed in duplicate and 3 independent experiments were analyzed. The results correspond to the means ± standard deviation (SD) of the duplicate reactions of three independent experiments. The relative gene expression was determined with the formula fold induction: 2^−ΔΔCt^, where ΔΔCt = (Ct GI [unknown sample] − Ct GI [reference sample]) − (Ct reference genes [unknown sample] − Ct reference genes [reference sample]). GI is the gene of interest. RS18 and RPL32 were used as internal controls. The reference sample is the MDA-MB-231 WT or shCtrl sample, chosen to represent 100% of the GI expression. The means ± SEM originated from 3 independent experiments realized in duplicates.

### 2.13. Tubule Formation

A growth-factor-reduced (GFR) Matrigel^®^ (Corning^®^, BD Biosciences, Franklin Lakes, NJ, USA) at 8.6 mg/mL was thawed on ice at 4 °C overnight before use. Ten microliters of GFR Matrigel^®^ were loaded into each well of a pre-cooled μ-Slide Angiogenesis plate, ibiTreat (ibidi^TM^, Martinsried, DE, USA), and the plate was incubated at 37 °C for 30 min. As mentioned in the Materials and Methods section, 1.5 × 10^4^ GFP-HUVECs cells were seeded in 50 μL of TCM to be tested and for controls, EGM-2, EBM-2, and 0.8% FBS DMEM. The plate was then incubated at 37 °C in a humid atmosphere in the presence of 5% CO_2_ for 8 h. A photography of each well was taken using a fluorescence microscope (X4) coupled to a camera. After 8 h at 37 °C, the cells were imaged at ×4 magnification on a Nikon eclipse 300 inverted microscope. The total network length and branching number were assessed using AutoTube [24]. The results are the means of random fields in 3 replicates and were repeated three times.

### 2.14. Endothelial Proliferation and Migration

An MTT assay was realized as described in [25]. Briefly, HUVECs were seeded in 96-well plates at a density of 1 × 10^4^ cells/mL in 100 μL of growth medium. Twenty-four hours later, the medium was replaced by 100 μL of TCM to be tested or control conditions (EGM-2, EBM-2 and 1% FBS DMEM) after rinsing the cells with PBS. Then, 20 μL of MTT (5 mg/mL) were added into each well after 0, 24, 48, and 72 h of treatment. Four hours later, 150 μL of dimethyl sulfoxide were added to each well. The absorbance (optical density, OD) at 560 nm was measured using a microplate reader (Tecan Infinite^®^, Mannedorf, Switzerland). The experiments were performed in triplicate. Migration experiments were carried out using ThinCert^TM^ cell culture inserts (BD Biosciences, Franklin Lakes, NJ, USA) in 8-µm-pore, fibronectin-coated membranes in a 24-well plate, as described in [26]. Briefly, HUVECs were seeded at a density of 0.15 × 10^6^ cells/cm^2^ on ThinCert^TM^ pre-coated with fibronectin from bovine plasma (Sigma-Aldrich, Saint-Louis, MI, USA) at 7 μg/mL overnight. The surplus was eliminated. After 30 min of hood drying, the lower well was filled with 800 μL of EGM-2, EBM-2, 0.8% FBS DMEM, and 48 h TCM to be tested containing 182 μL of fresh DMEM 3.5% FBS (for a final FBS concentration of 0.8%). Two hundred microliters of the HUVEC cell solution adjusted to 5 × 10^4^ cells/mL in EBM-2 were added to the upper well of each insert. The 24 well-plates were incubated at 37 °C in a humid atmosphere in the presence of 5% CO_2_. After 8 h, the medium was removed and replaced with cold methanol for 15 min at RT to fix the cells. The inserts were then rinsed by successive baths in distilled water. The cells that did not migrate on the upper well of the insert were eliminated using a cotton swab. The membranes were excised from inserts and mounted on microscopic observation slides with a ProLong^®^ Gold Antifade Reagent mounting medium (with DAPI (4′6-diamidino-2-phenvlindole)) (Invitrogen, Waltham, MA, USA). The cells were counted on 9 random microscopic fields per membrane using a fluorescence microscope (X20) (Evos, Thermo Fisher Scientific, Waltham, MA, USA) coupled to a camera. The experiments were carried out in triplicate and repeated with three independent TCM.

### 2.15. Proteomics 

For label-free quantitative proteomics, three independent biological replicates on secretome extracts for shLRP-1 and shRNA-control cell lines have been performed. Ten micrograms of proteins were loaded on a 10% acrylamide SDS-PAGE gel, and the proteins were visualized by Colloidal Blue staining. The migration was stopped when the samples had just entered the resolving gel, and the unresolved region of the gel was cut into only one segment. The steps of sample preparation and protein digestion by trypsin were performed as previously described [27]. A nanoLC-MS/MS analysis was performed using an Ultimate 3000 RSLC Nano-UPHLC system (Thermo Fisher Scientific, Waltham, MA, USA) coupled to a nanospray Orbitrap Fusion^TM^ Lumos^TM^ Tribrid^TM^ Mass Spectrometer (Thermo Fisher Scientific, Waltham, MA, USA). Each peptide extract was loaded on a 300-μm ID × 5 mm PepMap C18 precolumn (Thermo Fisher Scientific, Waltham, MA, USA) at a flow rate of 10 μL/min. After a 3-min desalting step, the peptides were separated on a 50-cm EasySpray column (75 μm ID, 2 μm C18 beads, 100 Å pore size, ES803A rev.2, Thermo Fisher Scientific, Waltham, MA, USA) with a 4–40% linear gradient of solvent B (0.1% formic acid in 80% ACN) in 115 min. The separation flow rate was set at 300 nL/min. The mass spectrometer operated in positive ion mode at a 2.0 kV needle voltage. The data were acquired using the Xcalibur 4.1 software in a data-dependent mode. MS scans (m/z 375–1500) were recorded at a resolution of R = 120,000 (@ m/z 200) and an AGC target of 4 × 10^5^ ions collected within 50 ms, followed by a top speed duty cycle of up to 3 s for MS/MS acquisition. Precursor ions (2 to 7 charge states) were isolated in the quadrupole with a mass window of 1.6 Th and fragmented with HCD@30% normalized collision energy. MS/MS data were acquired in the ion trap with the rapid scan mode, an AGC target of 3 × 10^3^ ions, and a maximum injection time of 300 ms. The selected precursors were excluded for 60 s. Protein identification and Label-Free Quantification (LFQ) were done in Proteome Discoverer 2.4. The MS Amanda 2.0, Sequest HT, and Mascot 2.4 algorithms were used for protein identification in batch mode by searching against a Uniprot Homo sapiens database (75,093 entries, release 20 May 2020). Two missed enzyme cleavages were allowed for trypsin. Mass tolerances in MS and MS/MS were set to 10 ppm and 0.6 Da. Oxidation (M), acetylation (K), and deamidation (N, Q) were searched as dynamic modifications, and carbamidomethylation (C) as a static modification. Peptide validation was performed using the Percolator algorithm [28], and only “high confidence” peptides were retained, corresponding to a 1% false discovery rate at the peptide level. A Minora feature detector node (LFQ) was used along with the feature mapper and precursor ion quantifier. The normalization parameters were selected as follows: (1) Unique peptides, (2) Precursor abundance based on intensity, (3) Normalization mode: total peptide amount, (4) Protein abundance calculation: summed abundances, (5) Protein ratio calculation: pairwise ratio-based, and (6) Hypothesis test: *t*-test (background-based). Quantitative data were considered for master proteins, quantified by a minimum of 2 unique peptides, fold changes above 2, and a statistical *p*-value lower than 0.05. The mass spectrometry proteomics data have been deposited in the ProteomeXchange Consortium via the PRIDE [29] partner repository with the dataset identifier PXD022978.

### 2.16. S-2251^TM^ Assay 

A chromogenic substrate selective for plasmin, S-2251 (Chromogenix, Diapharma, Westchester, NY, USA), was used to follow the initial rate of plasminogen activation by measuring p-nitroaniline generation. TCM were mixed with a buffer containing 0.1 M Tris-HCl pH 7.8 and 20 µL of plasminogen EACA reconstituted at 10 U/mL (Calbiochem, Darmstadt, Germany). To initiate the reaction, 20 µL of 3.5 mM S-2251^TM^ were added to each well. The generation of plasmin was detected by measuring the absorbance of the p-nitroaniline release every 30 min at 405 nm during 10 h (Tecan Infinite^®^, Mannedorf, Switzerland).

### 2.17. Patient Tumor-Derived Breast Cancer Xenografts (PDX)

Tumor fragments used to generate PDXs were collected from patients upon signing an informed consent, and PDX models have been generated according to previous studies [30].

### 2.18. RNA Seq

The RNA seq analysis was outsourced either to Integragen (France) or Novogene (China) by using poly-T oligo enriched RNA, a strand-specific library, and paired-end sequencing (Illumina).

### 2.19. Statistical Analysis 

The data are expressed as means +/− SEM, as indicated in the figure legends. For the statistical analysis, an independent t-test in vitro and a non-parametric Mann–Whitney in vivo were used to assess the significance of the mean differences. The differences were considered significant at a *p* value of 0.05 or less. * *p* < 0.05; ** *p* < 0.01; *** *p* < 0.005; **** *p* < 0.001. 

## 3. Results

### 3.1. LRP-1 Is Preferentially Expressed in TNBC Cell Lines

Using RNA-sequencing (Xentech biotechnology), we analyzed the quantity and the sequences of LRP-1 RNA in xenograft (PDX) derived from 20 breast cancer patients, including 12 TNBC and 8 non-TNBC (seven luminal and one HER2+). We highlighted no significant differences between the two groups but a trend of higher LRP-1 RNA expression in the TNBC group. However, LRP-1 RNA expression was found to be higher in 8/12 of TNBC PDXs compared to the average expression of the non-TNBC PDXs (with a mean of 67.86 vs. 23.07) (Figure 1A). We also evaluated the LRP-1 expression level in TNBC cell lines, MDA-MB-231, Hs-578T, BT-20, and 4T1, and in non-TNBC cell lines, MCF-7, SKBR3, and T47D. LRP-1 was found to be more expressed at the transcriptional and translational levels in TNBC cell lines (MDA-MB-231 > 4T1 > Hs578T > BT-20) in comparison to non-TNBC cell lines (T47D > MCF-7 > SK-BR3) (Figure 1B,C). Therefore, to investigate LRP-1’s role in TNBC progression, we used the stably transfected MDA-MB-231 cell line to allow for a constitutive expression of LRP-1-targeting shRNA (shLRP-1) or a scrambled shRNA (shCtrl). RT-qPCR and the immunoblot showed a significant decrease in LRP-1 mRNA (by 60%) and protein (by 67%) expression, respectively, in shLRP-1 MDA-MB-231 cells compared with shCtrl (Figure 1D–F). These results validated our LRP-1 study model in MDA-MB-231 cells. As shown in Appendix A, the LRP-1 expression in MDA-MB-231 without antibiotic selection pression showed no significant difference up to 35 days, indicating that LRP-1-targeting shRNA was stable over time and compatible with in vivo experiments (Appendix A).

### 3.2. LRP-1 Acts as a Pro-Tumorigenic Receptor, by Modulating Tumor Angiogenesis, in an Orthotopic Mammary Fat Pad TNBC Model

To determine LRP-1’s exact role in the in vivo TNBC progression, we performed mammary fat pad experiments by injecting shLRP-1 or shCtrl MDA-MB-231 cells orthotopically into nude mice and followed the tumor development for 28 days. Significant tumor volume differences appeared 14 days post-injection. The volume of the shLRP-1 tumors was reduced by 63% compared with shCtrl (mean of 118.83 ± 64.04 vs. 323.43 ± 92.65 mm^3^; median of 90.32 vs. 323.7 mm^3^, *** *p* < 0.0001) (Figure 2A). Twenty-eight days after injection, three quarters of shCtrl tumors had reached the endpoint versus one sixth of shLRP-1 tumors (8/12 vs. 2/12 tumors). Tumor volume differences persisted on living mice and ended up reaching, after 28 days later, a 64% reduced tumor volume in shLRP-1 MDA-MB-231 tumors compared with shCtrl (mean of 507.32 ± 101.36 vs. 1399.30 ± 347.91 mm^3^; median of 508.54 vs. 1322.22 mm^3^; *** *p* < 0.001) (Figure 2A). To examine the in vivo functional aspects of neo-formed vascular networks within tumors, we used the Dynamic Contrast Enhancement (DCE)-MRI and Fluorescent Molecular Tomography (FMT) imaging methods. As shown in Figure 2B, the temporal changes in contrast enhancement due to the gadolinium (Clariscan^®^) concentration within tumors after an intravenous bolus injection allowed us to observe fully perfused shCtrl tumors, while shLRP-1 tumors appeared only superficially perfused for a quarter of their circumference. To keep exploring the functional aspect of the vascular network, we used a long-circulating near-infrared fluorescent blood-pool agent (AngioSense^TM^-750). We observed a clear heterogeneity within tumor groups that did not allow us to conclude significantly on the slighter AngioSense^TM^-750 signal trend in shLRP-1 tumors compared to shCtrl (Figure 2C). However, the major population of shCtrl tumors [1] with an AngioSense^TM^-750 signal from 180 to 260 pmol presented a comparable signal on the tumors’ edges (Figure 2C, right panel). One of shCtrl tumors [2] stood out with a different profile and half the signal recovered (87 pmol) compared with the others. Concerning shLRP-1 tumors, we observed different profiles. From one low vascularized tumor with 38 pmol [5] to what seems to be a hyperpermeable marked profile with 269 pmol of AngioSense^TM^-750 signal [4]. Nevertheless, we found a major shLRP-1 population [3] with the same profile, characterized by an accumulation of AngioSense^TM^-750 from 111 pmol to 251 pmol in the heart of the tumor (Figure 2C, right panel). The results obtained using a HypoxiSense^TM^-680 fluorescent imaging agent that detects carbonic anhydrase 9 (CA IX) tumor cell surface expression revealed a rise of hypoxia in shLRP-1 tumors compared to shCtrl (0.079 ± 0.020 vs. 0.010 ± 0.04 pmol/mm^3^) (Figure 2D). Both LRP-1 immunoblots and RT-qPCR realized from tumors samples confirmed a LRP-1 protein repression of more than 50% (Figure 2E) and more than 70% at the transcriptional level (Figure 2F) at the end of the protocol in shLRP-1 tumors compared to shCtrl. CD31 labeling followed by a microvascular density (MVD) analysis were performed and highlighted differences of vascularization, revealed by a decrease of the vessel number in the shLRP-1 tumor section (−50% ± 7%, ** *p* < 0.01) (Figure 2G). In line with these observations, HES staining showed the largest necrosis areas in shLRP-1 tumors compared to shCtrl (52 ± 6% vs. 20 ± 4% of tumor area, ** *p* < 0.01) (Figure 2H). The count of mitoses did not reveal any difference between the two groups (Figure 2I). Thus, the vascular networks formed within shLRP-1 tumors presented morphological and functional differences compared to shCtrl, which were decisive for the primary tumor progression. This seems to be explained by the microenvironment‘s physicochemical properties modulation, especially hypoxia.

### 3.3. LRP-1 Repression Alters Angiogenesis in MDA-MB-231 Matrigel^®^ Plugs and CAMs Assays

To understand how LRP-1 repression in MDA-MB-231 cells may affect in vivo neo-angiogenesis, we performed a Matrigel^®^ plug (MP) assay while using DCE-MRI and FMT preclinical modalities to pull out information on vascular features inside the plugs. We used the AngioSense^TM^-680 agent in vivo at D7 and ex vivo at D21 in FMT after injecting tumor cells mixed with Matrigel^®^. As shown in Figure 3A,B, the fluorescence intensity was about 7-fold lower in vivo at D7 (22.7 ± 9.3 vs. 162.9 ± 46.9 pmol, ** *p* < 0.01) and ex vivo at D21 (0.7 ± 0.7 vs. 13.2 ± 2.2 pmol, * *p* < 0.05) in shLRP-1 MDA-MB-231 MPs compared to shCtrl. By using DCE-MRI, we showed that shLRP-1 MPs perfusion appeared less effective than in shCtrl (Figure 3C–E). Maximum intensity value analyses confirmed that shLRP-1 MPs were less perfused than shCtrl (1500 ± 108 vs. 1250 ± 73 A.U, *** *p* < 0.001), and the quantification of the area under the curve (AUC), which reflects the total amount of contrast transiting through the regional vascular system, highlighted a decreased perfusion in shLRP-1 MPs by 45% compared to shCtrl (3294 ± 237 vs. 1868 ± 217 A.U, ** *p* < 0.01). The MVD analysis revealed, similarly to the mammary fat pad experiment, a 40% decreased vessel number in shLRP-1 MPs compared to shCtrl (42 ± 3 vs. 28 ± 2 vessels/field, ** *p* < 0.01) (Figure 3F, middle and right panel). Additionally, we evaluated the angiogenic properties of LRP-1 expressed by MDA-MB-231 cells in ovo, using a chick embryo chorioallantoic membrane (CAM) assay [21]. Using a MATLAB™ homemade plugin, the segmentation of the angiogenesis showed that shLRP-1 CAMs grafted with shLRP-1 MDA-MB-231 cells showed a decreased neo-angiogenic vessel length (4606 ± 1021 vs. 2350 ± 439 pixels, * *p* < 0.05) and branching (71 ± 17 vs. 46 ± 12 pixels, * *p* < 0.05) compared with shCtrl (Figure 3G,H). In accordance with results obtained on tumor mammary fat pad, we also observed 1/3 of hemorrhagic CAMs when shLRP-1 MDA-MB-231 were grafted (Appendix A).

### 3.4. LRP-1-Down-Regulated MDA-MB-231 Secretome Modulates the Angiogenic Potential of Endothelial Cells

To explore how LRP-1 influences tumor progression and angiogenesis, we investigated whether a LRP-1-silenced MDA-MB-231 secretome could modulate the angiogenic potential of endothelial cells (ECs). The in vitro effects of shLRP-1 or shCtrl tumor conditioned media (TCM) were assessed on the migratory, proliferative capacities and tube formation abilities of HUVECs. The results on cell proliferation indicated that HUVECs were relatively more proliferative (+19 ± 4%, * *p* < 0.05) when incubated for at least 48 h in shLRP-1 MDA-MB-231 TCM compared with shCtrl (Figure 4A). As seen in Figure 4B,C, we showed that shLRP-1 MDA-MB-231 TCM were significatively less chemoattractant than shCtrl (Figure 4B). Indeed, we measured a significant 58% decrease in migrated HUVECs toward shLRP-1 TCM, compared with shCtrl (Figure 4C). Finally, ECs tubulogenesis assays revealed that HUVECs stimulated by shLRP-1 MDA-MB-231 TCM displayed decreased abilities to organize themselves into tubule structures compared to control conditions (Figure 4D). The segmentation of tubulogenesis revealed a significant decrease of the total length (15 ± 3.8%, *** *p* < 0.001) and branching number (+30 ± 1%, *** *p* < 0.001) in shLRP-1 TCM-stimulated conditions compared with shCtrl (Figure 4E).

### 3.5. MDA-MB-231 Secretome Analysis Reveals That LRP-1 Angiogenic Effects Involved TGF-β and Plasminogen/Plasmin Pathways

To decipher the mechanisms by which LRP-1 can influence tumor progression and angiogenesis, 24 h shLRP-1 and shCtrl cells secretomes were investigated using mass spectrometry-based proteomics. Intracellular proteins, most certainly coming from exosomes, were excluded. When LRP-1 is stably repressed in the cells, many factors (whether pro- or anti-angiogenics) are modulated, as shown on the representative heatmap (Figure 5A). Based on an in-depth analysis via the Proline software and using the GSEA and Ingenuity Pathways for pathway representation, we highlighted a preferential modulation scheme of certain pathways, such as the transforming growth factor-β (TGF-β) signaling (notably TGFβ-1, TGFβ-2, TGFβI) and the plasminogen/plasmin (PP) system (including PLG, PLAT, and a batch of SERPIN) (Figure 5B). In addition, TIMP-1, TIMP-2, and TIMP-3 with ratios of 35.37, 3.79, and 98.13, respectively, were enriched in a shLRP-1 secretome compared to shCtrl, as well as THBS1 with a ratio of 39.17 (Appendix A), suggesting a strong regulation of proteinase activity and anti-angiogenic effects. Pro-angiogenic molecules such as ECM1, GRN, and FST were also enriched with ratios of 77.49, 12.04, and 15.31, respectively (Appendix A). The modulation of the PP system was confirmed by measuring plasmin activity using S-2251^TM^ (HD-Val-Leu-Lys (pNA)) (Figure 6). The photometric measurements of plasmin activity demonstrated an exponential increase in plasmin activity in shCtrl MDA-MB-231 TCM, reaching an optical density at 405 nm (OD405) of 2.70 ± 0.1 after 630 min. In contrast, a slower conversion of plasminogen into plasmin was measured in shLRP-1 MDA-MB-231 TCM with an OD405 of 1.70 ± 0.02 after 630 min (Figure 6A). The data obtained from 24 h HUVEC-conditioned media by shLRP-1 or shCtrl TCM showed more pronounced effects (Figure 6B). Thus, HUVECs stimulated by shLRP-1 TCM exhibit a decreased plasmin activity compared to HUVECs stimulated by shCtrl, leading to a lesser propensity to migrate and invade.

## 4. Discussion

This study aimed to clarify LRP-1’s role in TNBC tumor growth, and more precisely its involvement in tumor angiogenesis using an MDA-MB-231 cell line-based model. Previous data from our group and others have shown that the expression of LRP-1 and LDLR was higher in mammary tumor tissues [31,32], contributing to LDL-C uptake from the blood [33] and a poor prognosis [34]. In particular, LRP-1 was shown to be involved in the invasiveness of luminal and TNBC subtypes of BC [18,35,36]. In recent years, LRP-1 was shown to be involved in angiogenesis, notably by regulating LRP1-dependent signaling pathways in different endothelial processes, such as proliferation, migration, permeability, and tube formation [37,38]. LRP-1 also plays an essential role in vascular homeostasis, by having a protective role in atherosclerosis pathogenesis and aneurysm formation [39]. Furthermore, some studies have revealed the angio-modulatory capacities of LRP family members in various solid tumors, including BC [40,41,42]. A PDX analysis comparing the LRP-1 RNA expression of TNBC versus non-TNBC showed no significant results, in line with the searched databases. This could be a consequence of the inherent heterogeneity of this aggressive subtype [43]. However, 3/4 of TNBC PDXs we had access to have a higher expression than the average non-TNBC PDXs. Therefore, the study of the role of LRP-1 appears to be relevant for a majority of TNBC. Moreover, a more accurate TNBC subtyping of the PDXs—such as a basal-like or non-basal-like distinguo—could show potential correlations with LRP-1 expression.

Here, we showed that LRP-1 plays a more decisive role, not only by contributing to cell survival and proliferation [44]; it modulates (directly or indirectly) the angiogenic balance through its pivotal roles within the tumor microenvironment. We showed that LRP-1 repression in MDA-MB-231 tumors led to a significant tumor growth decrease (64%) compared to the control group. The lower proliferative capacities of shLRP-1 cells observed in vitro (15–20%, data not shown) are not sufficient to explain such a difference in tumor volume. Otherwise, no significant difference in the mitotic index in the viable parts of the tumors was found.

As angiogenesis is required for tumor progression and growth [11], DCE-MRI experiments were conducted to assess tumor perfusion and enable the depiction of physiological alterations as well as morphological changes [45]. shLRP-1 tumors characterized by a decreased tumor perfusion in vivo exhibited numerous unsuccessful structures, displaying a CD31 signal but without lumen, suggesting that the stimulation of angiogenesis was present and sustained but unable to reach shCtrl vascular achievement. The in vivo vascular density evaluation in FMT confronted us with intra-tumor heterogeneity. Two major distinct populations were found according to the signal distribution—either peripheral tumors, in shCtrl, or central, in shLRP-1 tumors. An accumulation of fluorochrome in the peritumoral tissue is thought to be due to highly leaky vessels or a potential hemorrhage within tumors [46]. Certain CD31-stained shLRP-1 tumor sections exhibited large structures resembling hemorrhagic lakes rather than vessels, but anastomoses were also observed, highlighting a marked vascular anarchy when LRP-1 is repressed in MDA-MB-231. shLRP-1 tumors showed a significant increase in necrosis compared to shCtrl, as a direct result of the increased hypoxia. As LRP-1 is known to be upregulated by hypoxia [47], we ascertained that its expression was still low enough in our in vivo tumor model at the protocol end. As a common phenomenon in most malignant tumors, hypoxia leads to an advanced but dysfunctional vascularization, by inducing an imbalance between pro- and anti-angiogenic factor production, thus leading to a rapid and chaotic blood vessel formation increase [48].

By focusing on in vivo and in ovo angiogenic assays, we highlighted the LRP-1-silenced cells’ difficulties in supporting angiogenesis. The in vivo and ex vivo vascular densities found in MPs were indeed lower when LRP-1 expression was repressed, which is consistent with the decreased vessel numbers in CD31-stained MPs sections. In line with these data, blood perfusion appeared less efficient in shLRP-1 MPs. As for the CAMs assay, it demonstrates that the vascular networks generated by shLRP-1 cells exhibited a lesser overall length and a lower number of branchings. These results corroborate that LRP-1 plays a significant role in the outcome of angiogenic processes in MDA-MB-231 tumor cells.

In vitro, LRP-1 influences the tumor cells’ secretome which shapes EC behaviors among the microenvironment cells. We showed that the shLRP-1 MDA-MB-231 cells’ secretome decreases the angiogenic potential of HUVECs by impacting their ability to form a 3D-tubular network on Matrigel^®^ and, unsurprisingly, their migratory capacities. However, we found that shLRP-1 TCM led to a higher EC proliferative rate over time than shCtrl. The overall growth of a vasculature is the result of both proliferation and migration controlled by a myriad of factors in the tumor microenvironment, including many pro- and anti-angiogenic factors [49]. In a computational model, the authors have modulated proliferation and migration rates separately. They have demonstrated that an EC proliferation increase at the expense of migration leads to an increase in sprouts, which then mostly exhibit anastomoses preventing vessel functionality [50]. 

Through a proteomic approach, we demonstrated the extensive LRP-1’s influence on the MDA-MB-231 tumor cells’ secretome, where 962 proteins were identified. When it comes to identifying by which precise molecular pathways LRP-1 plays its part on tumor progression and angiogenesis, the task is intricate. We highlighted a solid modulation of TGF-β signaling as well as a modulation of the plasminogen/plasmin (PP) system. Under physiological conditions and in early stages of carcinogenesis, TGF-β acts as a tumor suppressor by restricting cell growth and stimulating apoptosis to maintain homeostasis in the tissues. However, in advanced tumors, cancer cells escape TGF-β’s initial suppressive effects and use its regulatory functions to promote their progression with clear roles in processes supporting cancer cell invasion, epithelial-mesenchymal transition (EMT), immune response suppression, angiogenesis, and metastasis [51]. In addition, TGF-β contributes to matrix remodeling by increasing the expression of MMPs [52] and plasmin, creating a permissive environment allowing cancer cells to metastasize [53]. Through endogenous TGF-β1 activation, it has been shown that thrombospondin-1 (THBS1) up-regulates the PP system and promotes tumor cell invasion in MDA-MB-231 cells [54]. THBS1, over-expressed in LRP-1-repressed MDA-MB-231, is established as an anti-angiogenic and anti-tumoral protein [55]. Notably, THBS1 binds to free and cell-associated VEGF [56], and THBS1/VEGF complexes are internalized via LRP-1 [57], suggesting that LRP-1 contributes to VEGF bioavailability during neovascularization. No clear modulation of VEGF by LRP-1 could be demonstrated, as no significant difference in VEGF-immunostained tumor sections was measured. However, an increase in VEGF transcriptional expression in tumors has been shown, certainly in response to hypoxia, because this increase was not measured in vitro (data not shown). In THBS1 up-regulated cells, the secreted VEGF could be sequestrated, and is thus not sufficient for the cells to ensure a proper VEGF-stimulated angiogenesis. As THBS1 regulates vessel stabilization, its overexpression has been shown to suppress vascular growth and expand vessel diameter [58], suggesting that it could be associated with dysfunctional angiogenesis, like in Fabry disease [59]. Despite an increased plasminogen expression and one of its activators in shLRP-1 TCM, a decreased plasmin activity was measured. The explanation appears more sophisticated than the unavailability of plasminogen or its activators, suggesting the involvement of system inhibitors such as SERPINE1/2 (PAI-1/2) or SERPINC1 (antithrombin-III), able to thwart the enzymatic cascade [60]. Angiogenesis is associated with an important extracellular remodeling involving different proteolytic systems, among which the PP system plays an essential role. EC migration is associated with significant proteolysis upregulation, and, conversely, PP system inhibition reduces angiogenesis in vitro [61]. Thus, the prevention of in vitro HUVECs’ tubular structure formation in shLRP-1 TCM is consistent with the decreased plasmin activity in HUVECs CM after shLRP-1 TCM stimulation, given that pseudotube formation is based on ECs’ proteolytic activity and migratory capacities generated in response to their environment. However, genetically altered mice for the PP system developed without overt vascular anomalies, indicating a possible compensation by other proteases in vivo [61]. Furthermore, SERPINF1, expressed five times more in shLRP-1 TCM, has been described as an inhibitor of hypoxia-induced angiogenesis by either directly targeting HIF-1 or regulating HIF-1’s target genes signaling cascades, thus blocking EC survival, proliferation, and migration or leading to their apoptosis [62].

Although we have previously shown that shLRP-1 cells revealed an increased cell rigidity in vitro, with the drop in membrane extension dynamics directly reflecting their altered migratory capacities [19], these results could be divergent in vivo. When we set an experimental configuration that mimics the in vivo environment or approaches it, whether it is a CAMs assay or the formation of 3D spheroids, shLRP-1 cells grafts or spheroids exhibit a more invasive profile than expected compared to shCtrl (Appendix A). As hypoxia contributes to TGB-β up-regulation and EMT phenotype acquisition, resulting in cell mobility and metastasis, it could be the trigger of invasiveness in vivo. Moreover, a long exposure to hypoxia is associated with DNA breaks and a high frequency of replication errors, potentially leading to genetic instability and mutagenesis [63], and increasing the metastatic potential. A hypoxic environment, unfavorable to cell proliferation and survival, participates in the selection of cell clones that have acquired insensitivity to oxygen and nutrient deprivation [48]. In particular, MDA-MB-231 cells have been shown to secrete heat shock protein 90 alpha (eHsp90α) to mediate their survival under hypoxia [64]. The integration of such survival signals, leading to the epithelial-to-mesenchymal transition and migration in breast cancer cells, is dependent on the LRP-1 receptor [65]. Although the expression of Hsp90α was not identified in our analysis, it should nonetheless be excluded from future investigations, given its direct link with LRP-1 and the potential for its inhibition in TNBC [66]. As a mecanosensor of the tumor microenvironment, LRP-1 temporal expression during tumorigenesis could modulate the sensitivity of cells in response to stresses such as hypoxia. Thus, the question of whether LRP-1-repressed cells, less proliferative, with lower migratory properties in vitro, and forming primary tumors of smaller sizes in vivo, could surpass shCtrl MDA-MB-231 cells’ aggressiveness in the late tumorigenesis stages due to the hypoxia rise and a permissive signaling such as TGF-β is more than relevant and will be addressed later.

## 5. Conclusions

In the present study, we showed that LRP-1 emerges as an important matricellular player in the control of cancer-signaling events such as angiogenesis, by supporting tumor vascular organization in a way that appears dispensable but that is ultimately essential for the vascular effectiveness for tumor growth.

## Figures and Tables

**Figure 1 biomedicines-09-01430-f001:**
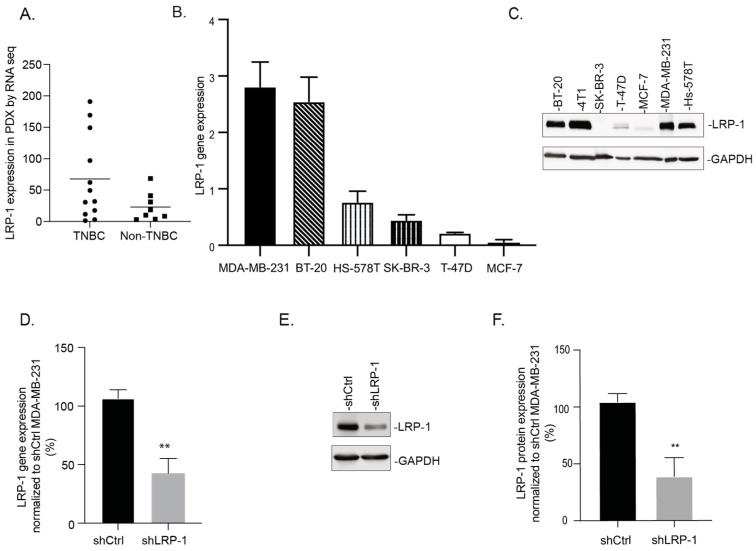
LRP-1 is preferentially expressed in TNBC cell lines. (**A**) LRP-1 RNA-sequencing in human breast cancer Patient Derived Xenograft (PDX). (**B**) mRNA levels in human breast cancer cell lines (MDA-MB-231, BT-20, Hs-578T, SK-BR-3, T-47D, MCF-7) analyzed by qRT-PCR (*n* = 3). (**C**) Representative immunoblot screening of LRP-1 expression in breast cancer cell lines (BT-20, 4T1, SK-BR-3, T-47D, MCF-7, MDA-MB-231, Hs-578T). (**D**) LRP-1 mRNA relative expression in shCtrl and shLRP-1 MDA-MB-231 cells determined by RT-qPCR and normalized to shCtrl MDA-MB-231 (*n* = 3). (**E**) Representative immunoblot of LRP-1 expression in shLRP-1 and shCtrl MDA-MB-231 cells expression. (**F**) Densitometric analysis of LRP-1 expression and normalization to shCtrl MDA-MB-231 (*n* = 4). Data points are mean ± SEM. *n* ≥ 3; ** *p* < 0.01 (Student *t*-test).

**Figure 2 biomedicines-09-01430-f002:**
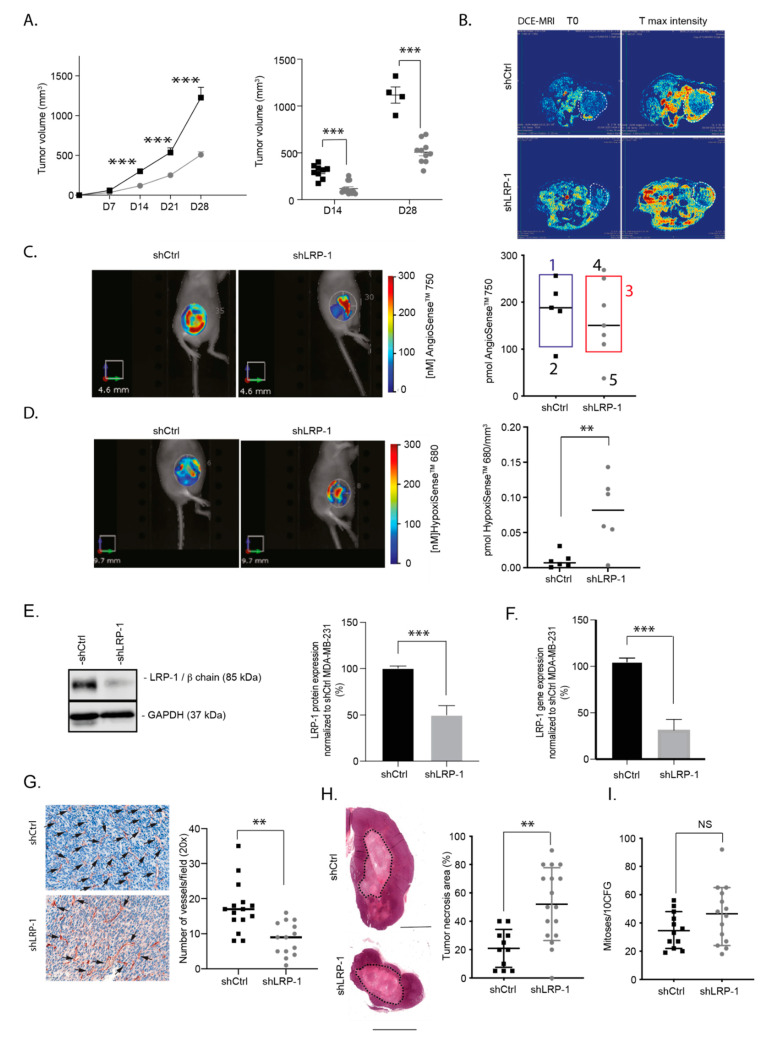
LRP-1 plays a pro-tumorigenic role in vivo, by supporting tumor angiogenesis, in MDA-MB-231 orthotopic xenografts. (**A**) (left panel) Tumor growth of shLRP-1 and shCtrl MDA-MB-231 orthotopic xenografts injected in BALBc/nu mice over 28 days (*n* = 12 per group). (right panel) Tumor volume repartition at D14 and D28 after implantation. (**B**) Images of T0 and Tmax intensity from DCE-MRI acquired in shLRP-1 and shCtrl-xenograft-bearing mice after an intravenous bolus injection of Clariscan^TM^. The dotted lines show the tumor area. (**C**) (left panel) Representative images of AngioSense^TM^-750 accumulation within BALBc/nu mice bearing shLRP-1 and shCtrl MDA-MB-231 xenografts. (right panel) 3D fluorescence quantification (pmol) (*n* = 5). Different populations have been identified according to imaging profiles (1,2,3,4,5). (**D**) (left panel) Representative images of HypoxiSense^TM^-680 accumulation within BALBc/nu mice bearing shLRP-1 and shCtrl xenografts. (right panel) 3D fluorescence quantification per tumor volume (pmol/mm^3^) (*n* = 6). (**E**) (left panel) Representative immunoblot of LRP-1 expression in harvested shLRP-1 and shCtrl MDA-MB-231 xenografts. (right panel) Densitometric analysis of LRP-1 expression and normalization to shCtrl xenograft expression (*n* = 3). (**F**) LRP-1 mRNA relative expression in harvested shLRP-1 and shCtrl MDA-MB-231 xenografts determined by qRT-PCR and normalized to shCtrl xenograft expression (*n* = 3). (**G**) (left panel) Representative microphotographs of CD31 immuno-localization on shLRP-1 and shCtrl MDA-MB-231 xenograft tissue sections (×200). Scale bar: 50 μm. (right panel) Number of vascular structures per 5 fields in CD31-stained sections of shLRP-1 and shCtrl MDA-MB-231 xenografts (×200) (*n* = 6). (**H**) (right panel) Representative HE-stained sections of shLRP-1 and shCtrl MDA-MB-231 xenografts. The dotted lines show the necrosis area. Scale bar: 500 μm. (left panel) Percentage of tumor necrosis area within shLRP-1 and shCtrl MDA-MB-231 xenograft sections (*n* = 11). (**I**) Number of mitoses per 10 high power fields (HPF) corresponding to 2 mm^2^ in shLRP-1 and shCtrl MDA-MB-231 xenograft tissue sections (×200) (*n* = 12). The data points are mean ± SEM. *n* ≥ 3; ** *p* < 0.01; *** *p* < 0.001 (Mann–Whitney or Student *t*-test).

**Figure 3 biomedicines-09-01430-f003:**
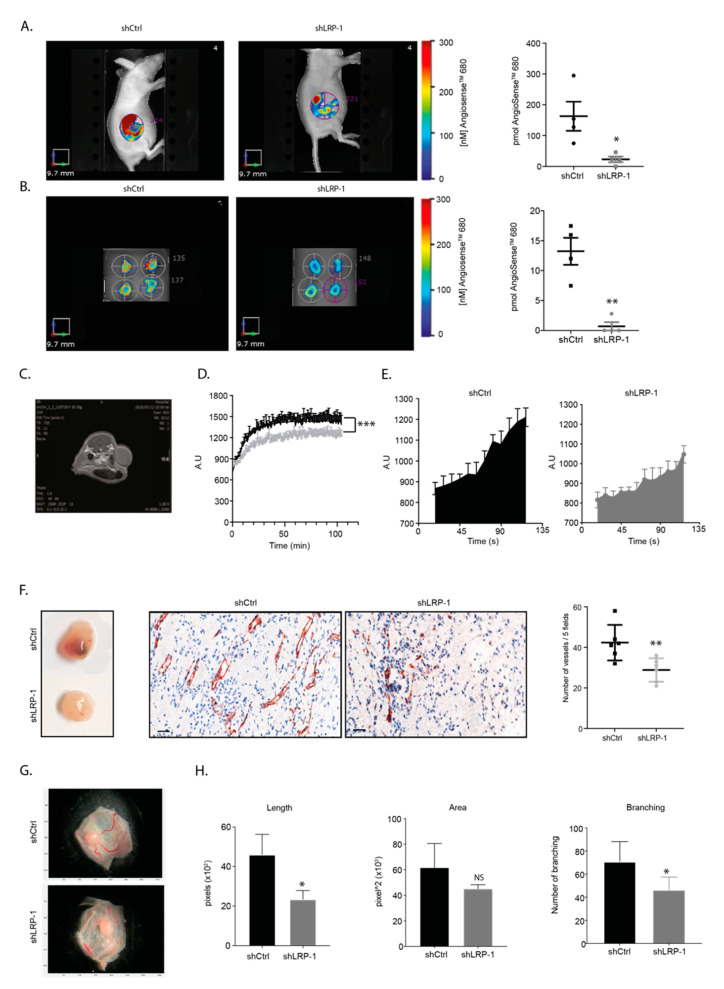
LRP-1 repression affects MDA-MB-231 Matrigel^®^ plug angiogenesis. (**A**) (left panel) Representative images of AngioSense^TM^-680 accumulation within BALBc/nu mice bearing shLRP-1 and shCtrl Matrigel^®^ Plug (MP). (right panel) 3D fluorescence quantification at day 7 (pmol) (*n* = 4). (**B**) (left panel) Representative images of AngioSense^TM^-680 accumulation in ex vivo harvested shLRP-1 and shCtrl MP. (right panel) 3D fluorescence quantification at the end of the protocol (pmol) (*n* = 4). (**C**) Coronal T1-weighted representative image of shCtrl MDA-MB-231 MP implantation. The dotted lines show the MP. (**D**) Representative DCE-MRI signal intensity curve acquired in shLRP-1 and shCtrl MDA-MB-231 MP-bearing mice after an intravenous bolus injection of Clariscan^TM^. A.U = Arbitrary Unit. (**E**) Representative Area Under the Curve (AUC) of shLRP-1 and shCtrl MDA-MB-231 MP. (**F**) (left panel) Representative macrophotographs of shLRP-1 and shCtrl MDA-MB-231 MP after excision. (middle panel) Representative microphotographs of CD31 immuno-localization on shLRP-1 and shCtrl MDA-MB-231 MP tissue sections (×200). Scale bar: 50 μm. (right panel) Number of vascular structures per 5 fields in shLRP-1 and shCtrl MDA-MB-231 MP tissue sections (×200) (*n* = 6). (**G**) Representative macrophotographs of shLRP-1 and shCtrl grafted CAMs and vascular segmentation using a homemade MATLAB© plugin. (**H**) Total length of the vascular structures, area, and number of branching quantified through pixel measurements in shLRP-1 and shCtrl-grafted CAMs (*n* = 5). The data points are mean ± SEM. *n* ≥ 3. * *p* < 0.05; ** *p* < 0.01; *** *p* < 0.001 (Mann–Whitney or Student *t*-test).

**Figure 4 biomedicines-09-01430-f004:**
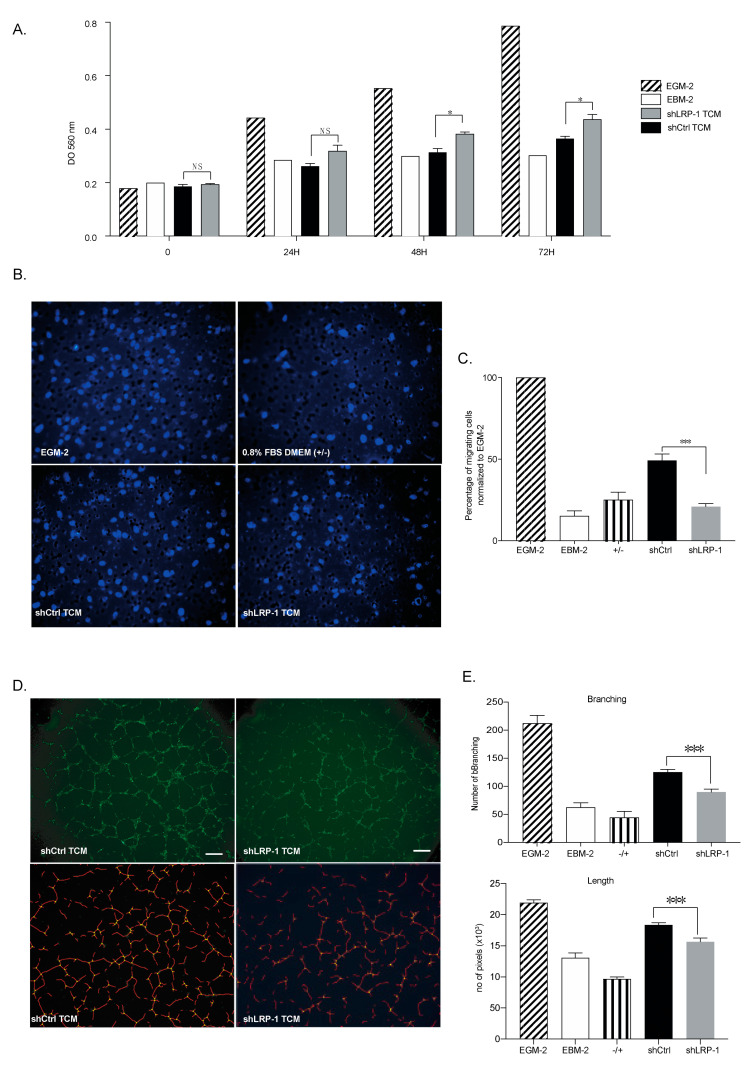
shLRP-1 MDA-MB-231 secretome restricts the angiogenic potential of endothelial cells. (**A**) MTT colorimetric cell proliferation assay of HUVECs incubated in EGM-2, EBM-2, shLRP-1, and shCtrl MDA-MB-231 TCM over time (0, 24 h, 48 h, and 72 h) (*n* = 3). (**B**) Representative microphotographs of migrating HUVECs through a fibronectin-coated 8-μm porous membrane by chemoattraction of EGM-2, 0.8% FBS DMEM (+/−), shLRP-1, and shCtrl MDA-MB-231 TCM for 8 h. (**C**) Histogram of the migrating cells’ percentage per condition normalized to EGM-2 values (*n* = 3). (**D**) (top panel) Representative microphotographs of HUVECs’ ability to form tubule-like structures when stimulated by shLRP-1 or shCtrl MDA-MB-231 TCM for 8 h. Scale bar: 75 µm. (bottom panel) Tubular-like structure segmentation using AutoTube Software [24] (**E**) (top panel) Number of branching and (bottom panel) surface area of tubular-like structures quantified through pixel measurements in EGM-2, EBM-2, 0.8% FBS DMEM (+/−), shLRP-1, and shCtrl MDA-MB-231 TCM conditions (*n* = 3). The data points are mean ± SEM. *n* ≥ 3. * *p* < 0.05; *** *p* < 0.001 (Student *t*-test).

**Figure 5 biomedicines-09-01430-f005:**
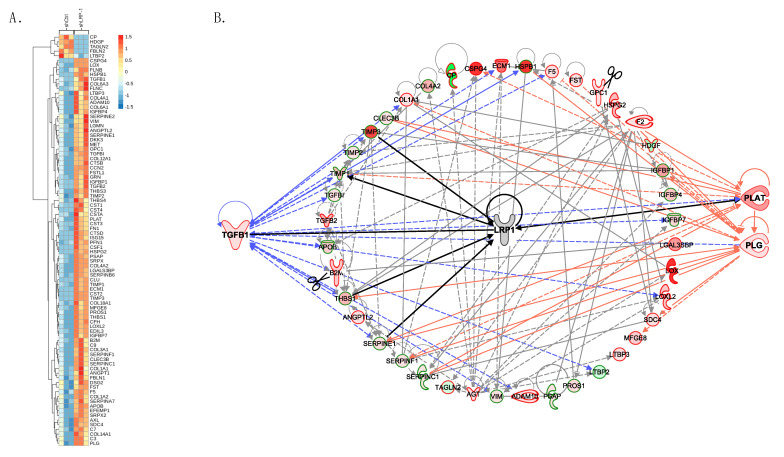
LRP-1 may affect angiogenesis through TGF-β signaling and the plasminogen/plasmin system modulation. (**A**) Colored heatmap generated from proteomics analysis data using the ggplot2 R package reflecting LRP-1’s influence in 24 h shLRP-1 and shCtrl MDA-MB-231 TCM. Comparison of proteomics profiles between shLRP-1 and shCtrl triplicate. Logarithmic scale of fold change from 1.5 to –1.5. (**B**) Representative pathway of LRP-1 modulations in MDA-MB-231 secretome. Among selective known genes linked to cancer progression and/or angiogenesis, protein–protein interactions were mapped using Ingenuity Pathways Analysis. TGF-β signaling (governed by TGF-β1, on the left) and the plasminogen/plasmin system (represented by PLG/PLAT, on the right) stand out for their privileged place within these multiple interactions organized around LRP-1.

**Figure 6 biomedicines-09-01430-f006:**
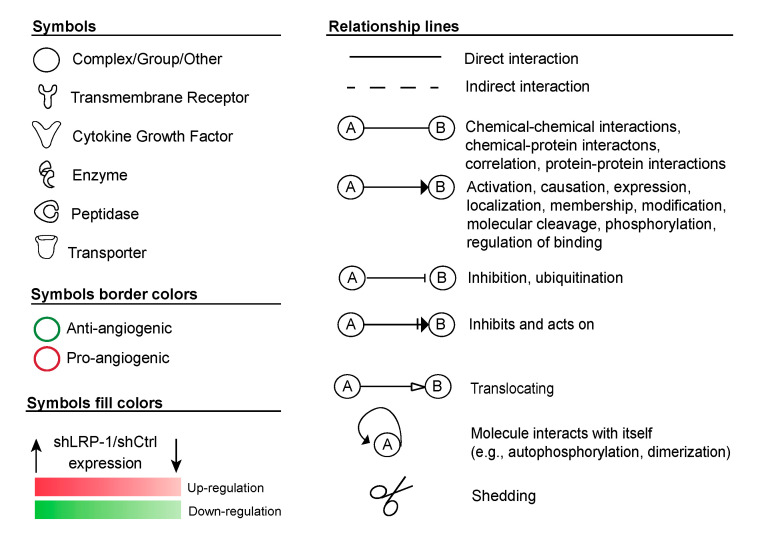
LRP-1 modulates plasmin activity in a 24-h MDA-MB-231 secretome. (**A**) the plasmin activity generated in 24 h shLRP-1 and shCtrl MDA-MB-231 TCM was measured every 30 min during 20 h by a colorimetric technique using a synthetic substrate of plasmin S-2251^TM^ (H-D-Val-Leu-Lys(pNA)) (*n* = 3). (**B**) the plasmin activity generated in 24 h HUVECs CM after their stimulation by 24 h shLRP-1 and shCtrl MDA-MB-231 TCM was measured every 30 min during 20 h by a colorimetric technique using a synthetic substrate of plasmin S-2251^TM^ (H-D-Val-Leu-Lys(pNA)) (*n* = 3). The data points are mean ± SEM. *n* ≥ 3. ** *p* < 0.01; *** *p* < 0.001 (Student *t*-test).

## Data Availability

The proteomics analysis revealed that LRP-1 supports tumor growth and angiogenesis through TGF-β signaling and the plasminogen/plasmin system. Concerning these last analyses, mass spectrometry proteomics data have been deposited into the ProteomeXchange Consortium (http://proteomecentral.proteomexchange.org (accessed on 15 August 2021)) via the PRIDE partner repository with the dataset identifier PXD022978.

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
