# Peer review of "LRP-1 Matricellular Receptor Involvement in Triple Negative Breast Cancer Tumor Angiogenesis"

_biomedicines, 2021, doi:10.3390/biomedicines9101430_

Round 1

Reviewer 1 Report

In this manuscript, the authors investigated the functional role of LRP-1 in the triple-negative breast cancer model MDA-MB- 21 231 cells. BY using in vitro and in vivo approaches, they demonstrated that LRP-1 has a role in the regulation of angiogenesis and tumor growth through the modulation of TGF-β signaling and plasminogen/plasmin system. The manuscript is well-written and the data support the authors' conclusions.

Here some minor comments:

3.1. LRP-1 is preferentially expressed in TNBC cell lines. Data from cancer atlas or other repositories should be used in support of this result. Moreover, this manuscript should be considered and discussed: Sci rep. 2016 Feb 5;6:20605. doi: 10.1038/srep20605.

Author Response

Dear Reviewer 1,

We greatly appreciate your comments on our submitted manuscript entitled “LRP-1 matricellular receptor involvement in triple negative breast cancer tumor angiogenesis (biomedicines-1354979). We have carefully revised the manuscript according to your suggestions, using red font to mark up the changes in our Word file. We have addressed specific points in the text below.

Reviewer 1

In this manuscript, the authors investigated the functional role of LRP-1 in the triple-negative breast cancer model MDA-MB- 21 231 cells. By using in vitro and in vivo approaches, they demonstrated that LRP-1 has a role in the regulation of angiogenesis and tumor growth through the modulation of TGF-β signaling and plasminogen/plasmin system. The manuscript is well-written and the data support the authors' conclusions.

Here some minor comments:

3.1. LRP-1 is preferentially expressed in TNBC cell lines. Data from cancer atlas or other repositories should be used in support of this result.

A: The Reviewer's remark is very pertinent. However, querying the databases unfortunately does not bring great results in our case. Indeed, few data bases are available specifically for TNBCs and their distinction within breast cancers is not always feasible. Nonetheless, we could provide some answers to the questions raised: We enriched our data with PDX thanks to the company Xentech with which we collaborate. Within the RNA sequencing of 123 available PDXs from all cancers, we identified 20 reliable breast cancers PDXs. When we classified them into two categories TNBC and non-TNBC (mainly comprising luminals and one HER2+), despite some inherent heterogeneity in the subtype leading to non-significative results, 3/4 of TNBC PDXs showed a higher expression than the average of non-TNBC. We have added this graph in Fig.1A and discussed it.

 Moreover, this manuscript should be considered and discussed: Sci rep. 2016 Feb 5;6:20605. doi: 10.1038/srep20605.

A: We thank The Reviewer for this relevant suggestion. We have added and discussed this publication, with others to supplement our discussion. 

We sincerely hope that this revised manuscript will be suitable for publication in Biomedicines.

Best regards,

J.DEVY

Reviewer 2 Report

In this investigation, the authors focus on the function of LRP1 expression in TNBC. They claim that LRP1 can modulate tumour growth via TGFbeta- and plasminogen signalling. This effect is basically attributed to vascularisation. As methods, MDA-MB-231 cells were investigated as an in-vitro system using LRP1 knock down, which has been established and published earlier. Additionally, these cells were used in a xenograft model. The authors used several up-to-date technologies such as imaging and mass spectroscopy for this project.

TNBC is still a challenge for the clinic. Any possible molecular target can therefore improve the therapy and understanding for this cancer type. The topic of this paper is therefore interesting and important.

The manuscript is generally well written. However, there are a few grammatical errors that should be eliminated before publication.

The major weakness of this study is the focus on one cell model only and the application of "loss of function" tests only for the mechanistical studies. The conclusions of this investigation would be greatly supported by extending the experiments to other TNBC cell-lines and including gain of function experiments.

mRNA was determined for a few cell lines representing major clinical BC subtypes. As there are databases present, which provide much more information on numerous BC cell lines these should be used for comparison. Such an analysis should also be done on patient data. Indeed, a quick look into GEPIA2 and METABRIC showed that LRP1 mRNA seems not specifically high-expressed in basal BC. Please include such data and discuss!

Minor points:

Line 129: It is not quite clear which concentration the authors mean. Was the dilution calculated from the original cell numbers or protein content or something else?

167 and others: I think matrigel is delivered with a given, lot-dependent concentration. This should be stated.

Line 244: Usually overnight incubations are done at 4°C – please check whether RT is correct.

Fig.1: It is useless to show control bars of 100% without showing standard deviation or standard error for these data. This holds true for other figures as well.

Author Response

Dear Reviewer 2,

We greatly appreciate your comments on our submitted manuscript entitled “LRP-1 matricellular receptor involvement in triple negative breast cancer tumor angiogenesis (biomedicines-1354979). We have carefully revised the manuscript according to your suggestions, using red font to mark up the changes in our Word file. We have addressed specific points in the text below.

Reviewer 2

In this investigation, the authors focus on the function of LRP1 expression in TNBC. They claim that LRP1 can modulate tumour growth via TGFbeta- and plasminogen signalling. This effect is basically attributed to vascularisation. As methods, MDA-MB-231 cells were investigated as an in-vitro system using LRP1 knock down, which has been established and published earlier. Additionally, these cells were used in a xenograft model. The authors used several up-to-date technologies such as imaging and mass spectroscopy for this project.

TNBC is still a challenge for the clinic. Any possible molecular target can therefore improve the therapy and understanding for this cancer type. The topic of this paper is therefore interesting and important.

The manuscript is generally well written. However, there are a few grammatical errors that should be eliminated before publication.

The major weakness of this study is the focus on one cell model only and the application of "loss of function" tests only for the mechanistical studies. The conclusions of this investigation would be greatly supported by extending the experiments to other TNBC cell-lines and including gain of function experiments.

A: We thank the reviewer for these interesting suggestions. It is true that the uniqueness of our model does not allow us to generalize to TNBC, so our statements were moderate. However, we would like to defend our position to use a cell line at this juncture. In view of the inherent heterogeneity of the TNBC subtype, we have chosen a TNBC cell line that expresses LRP-1 strongly to demonstrate the concept of the role that LRP-1 may play in TNBC. A role that may exist within TNBC, but that should not be generalized at this stage. The use of additional cell lines is part of our outlook to further study the role of the LRP-1 receptor in angiogenesis associated with TNBC. Regarding the chosen method of repressing the expression of the receptor, it is true, as reviewer 2 pointed out to us that the addition of expression in gain of function would have been very relevant. Unfortunately, such a method for a transmembrane receptor like LRP-1 is not feasible. We could possibly allow the ectopic expression of LRP-1 mini-receptors but not of the complete receptor, which would not fully answer the question asked and would induce potential interpretation bias since the mini-receptors would be able to interact with different ligands of LRP-1 without inducing cell signaling.

mRNA was determined for a few cell lines representing major clinical BC subtypes. As there are databases present, which provide much more information on numerous BC cell lines these should be used for comparison. Such an analysis should also be done on patient data. Indeed, a quick look into GEPIA2 and METABRIC showed that LRP1 mRNA seems not specifically high-expressed in basal BC. Please include such data and discuss!

A: The Reviewer's remark is very pertinent. However, querying the databases unfortunately does not bring great results in our case. Indeed, few data bases are available specifically for TNBCs and their distinction within breast cancers is not always feasible. Nonetheless, we could provide some answers to the questions raised: We enriched our data with PDX thanks to the company Xentech with which we collaborate. Within the RNA sequencing of 123 available PDXs from all cancers, we identified 20 reliable breast cancers PDXs. When we classified them into two categories TNBC and non-TNBC (mainly comprising luminals and one HER2+), despite some inherent heterogeneity in the subtype leading to non-significative results, 3/4 of TNBC PDXs showed a higher expression than the average of non-TNBC. We have added this graph in Fig.1A and discussed it.

Minor points:

Line 129: It is not quite clear which concentration the authors mean. Was the dilution calculated from the original cell numbers or protein content or something else?

A: For tumor conditioned media preparation, we do not give a numerical concentration because we have chosen to rely on the same number of cells to achieve a pair of conditioned media shCtrl and shLRP-1, to be compared only with each other. Tumor conditioned media are equivalent by pair, however the cellular concentrations to achieve them can slightly change from 0.8 to 1.2 million cells/mL from one pair of TCM to another. We have added this information into the Materials & Methods section.

167 and others: I think matrigel is delivered with a given, lot-dependent concentration. This should be stated.

A: We thank the Reviewer for this relevant comment, and we have added the used Matrigel concentration to the Materials & Methods section.

Line 244: Usually overnight incubations are done at 4°C – please check whether RT is correct.

A: We thank the Reviewer for spotted this error that we have corrected in the Materials and Methods section.

Fig.1: It is useless to show control bars of 100% without showing standard deviation or standard error for these data. This holds true for other figures as well.

A : Figures concerned have been modified.

We sincerely hope that this revised manuscript will be suitable for publication in Biomedicines.

Best regards,

J.DEVY

Round 2

Reviewer 2 Report

The authors have submitted a revised version of their manuscript. I think they answered sufficiently to my questions and also added addtiional data. So from my point of view, the manuscript can now be published